# Quantum majorization and a complete set of entropic conditions for quantum thermodynamics

Gilad Gour[1,2], David Jennings[3,4], Francesco Buscemi[5], Runyao Duan[6,7] & Iman Marvian [8]

What does it mean for one quantum process to be more disordered than another? Interestingly, this apparently abstract question arises naturally in a wide range of areas such as information theory, thermodynamics, quantum reference frames, and the resource theory of asymmetry. Here we use a quantum-mechanical generalization of majorization to develop a framework for answering this question, in terms of single-shot entropies, or equivalently, in terms of semi-definite programs. We also investigate some of the applications of this framework, and remarkably find that, in the context of quantum thermodynamics it provides the first complete set of necessary and sufficient conditions for arbitrary quantum state transformations under thermodynamic processes, which rigorously accounts for quantum-mechanical properties, such as coherence. Our framework of generalized thermal processes extends thermal operations, and is based on natural physical principles, namely, energy conservation, the existence of equilibrium states, and the requirement that quantum coherence be accounted for thermodynamically.

[1] Department of Mathematics and Statistics, University of Calgary, Calgary, AB T2N 1N4, Canada. [2] Institute for Quantum Science and Technology, University of Calgary, Calgary, AB T2N 1N4, Canada. [3] Department of Physics, University of Oxford, Oxford OX1 3PU, UK. [4] Department of Physics, Imperial College London, London SW7 2AZ, UK. [5] Department of Mathematical Informatics, Nagoya University, Chikusa-ku, Nagoya 464-8601, Japan. [6] Institute for Quantum Computing, Baidu Inc., 100193 Beijing, China. [7] Centre for Quantum Software and Information, Faculty of Engineering and Information Technology, University of Technology Sydney, Sydney, NSW 2007, Australia. [8] Departments of Physics & Electrical and Computer Engineering, Duke University, Durham, NC 27708, USA. Correspondence and requests for materials should be addressed to G.G. (email: gour@ucalgary.ca)

I rreversibility—the loss of order and the increase of disorder—is a fundamental and ubiquitous feature of physics that is typically described through thermodynamics and thermodynamic entropy. However, its scope goes above and beyond what one would ordinarily consider thermodynamic in nature. For example, the use of quantum entanglement within photonic quantum computing is subject to a form of irreversibility that need not be attached to either a particular energy scale or an equilibrium environment. Increasingly, a broader notion of irreversibility has been developed, that has been shown to include thermodynamic irreversibility as a special case, and has also allowed us to study intrinsically quantum mechanical order (such as entanglement or coherence) in contrast to classically ordered systems. Majorization is at the core of this development.

Majorization is a fundamental tool that finds application across a wide range of subjects from economics and statistics, to physics, chemistry, and pure mathematics[1]. At its core lies a notion of "deviations from uniformity", and the theory ties together mathematical techniques in convexity, combinatorics, and mathematical statistics.

An example of its use is in statistical mechanics of a physical system with $N$ energy levels. If we assume, for the sake of discussion, that the system is fully degenerate in energy, its thermal equilibrium state is described by the uniform probability distribution $\boldsymbol{\gamma} = \left(\frac{1}{N}, \dots, \frac{1}{N}\right)$ over the energy levels. Given any two other probability distributions $\mathbf{p} = (p_1, \dots, p_N)$ and $\mathbf{q} = (q_1, \dots, q_N)$, one might wish to say whether one is more or less out of equilibrium than the other. Majorization provides a concrete way of stating this. The distribution $\mathbf{p}$ is more ordered than $\mathbf{q}$ (or "$\mathbf{p}$ majorizes $\mathbf{q}$", written $\mathbf{q} \prec \mathbf{p}$) if $\mathbf{q} = D\mathbf{p}$ for some doubly stochastic matrix $D$[1]. A crucial property of majorization is that it can be equivalently formulated in terms of a complete set of monotones. For example, it is well-known that $\mathbf{q} \prec \mathbf{p}$ if and only if $\sum_k f(p_k) \geq \sum_k f(q_k)$ for all continuous real-valued convex functions $f$. Therefore, the value of any continuous convex function $f$ on statistical distributions can never increase under doubly stochastic transformations. Such functions are therefore monotones that quantify the deviation from equilibrium; moreover, they constitute a complete set of monotones because the comparison of their values provides a sufficient condition for the existence of a doubly stochastic transformation.

Majorization also finds extensive use in various parts of quantum information theory, such as in entanglement theory[2] and recent formulations of resource theories[3]. In particular, it has a central role in the recent thermodynamic frameworks using the quantum information theory[3–14]. In particular, it was found that state transformations with zero coherences in energy are fully characterized by thermo-majorization[5] (see also earlier works[15,16]), which is a natural generalization of majorization[4,17]. However, it was shown in ref. [10] that such thermo-majorization results are insufficient for describing quantum coherence under thermal operations, and that novel coherence measures are required. Low temperature coherence regimes were shown to admit solvable analysis[9], general coherence bounds were developed[11], and a framework for coherence based on the concept of asymmetry under time-translations was proposed[10,12]. However, a complete specification of the structure of non-equilibrium quantum states was still lacking.

A natural question is therefore whether there exists a generalization of majorization (or thermo-majorization) that can accommodate such intrinsically quantum-mechanical orderings. Several candidate generalizations exist[1,18,19], however, the one most relevant to our present work is called matrix majorization[16], which is a specialization to linear algebra of ideas coming from the theory of statistical comparison (see ref. [20] and references therein). Given two matrices of real numbers $A$ and $B$, we say that

$A$ matrix-majorizes $B$, and write $B \prec_m A$, if and only if $B = AX$ for some row stochastic matrix $X$. It is easy to see that this is a generalization of majorization: for the two-row matrices $A = \begin{bmatrix} \mathbf{p} \\ \mathbf{e} \end{bmatrix}$ and $B = \begin{bmatrix} \mathbf{q} \\ \mathbf{e} \end{bmatrix}$, with $\mathbf{e} \equiv (1, 1, \dots, 1)$, the relation $B \prec_m A$ is equivalent to $\mathbf{q} \prec \mathbf{p}$. Similarly, other variants of majorization, like thermo-majorization, are special cases of matrix majorization. However, such an ordering is inherently classical, being ultimately based on stochasticity, as opposed to coherent quantum processes.

A key component of our work is to generalize matrix majorization in a natural way into the quantum-mechanical setting, and to provide applications to a number of topics. Our first contribution to this is to provide a complete entropic description of a fully quantum-mechanical form of majorization. We then outline the core features of the solution and discuss the inclusion of quantum-mechanical symmetries. Our final contribution is to define a natural framework for quantum thermodynamics that is based on three physical assumptions, provide a complete set of entropic conditions and discuss limiting thermodynamic regimes of the theory.

## Results

**Definition of quantum majorization.** Our generalization of matrix majorization, which we call quantum majorization, defines a relation on bipartite quantum states, and consequently, due to the channel-state duality property of quantum theory, also defines a relation on quantum processes, i.e., completely positive and trace-preserving (CPTP) maps. Notice that notions equivalent to quantum majorization have previously been considered in refs. [21–25] in the contexts of quantum statistics and quantum information theory.

Definition 1: Let $\rho^{AB} \in \mathcal{B}(\mathcal{H}_A \otimes \mathcal{H}_B)$ and $\sigma^{AC} \in \mathcal{B}(\mathcal{H}_A \otimes \mathcal{H}_C)$ be two bipartite quantum states. We say that $\rho^{AB}$ quantum majorizes $\sigma^{AC}$, and write $\sigma^{AC} \prec_q \rho^{AB}$, if and only if there exists a CPTP map $\mathcal{E} : \mathcal{B}(\mathcal{H}_B) \to \mathcal{B}(\mathcal{H}_C)$ such that $\mathrm{id} \otimes \mathcal{E}(\rho^{AB}) = \sigma^{AC}$.

Remark 1: The preorder $\sigma^{AC} \prec_q \rho^{AB}$ is not symmetric with respect to the action of $\varepsilon$. It means that $\rho^{AB}$ quantum majorizes $\sigma^{AC}$ on $B$. However, in the remaining of this paper, it will be clear from the text that the action of $\mathcal{E}$ is on system $B$.

It is clear from Definition 1 that $\rho^A = \sigma^A$ is a necessary condition, called the compatibility condition, for the ordering of states to hold since $\mathcal{E}$ is trace-preserving, and when it holds the two states are said to be compatible. Moreover, in the special case that the marginals satisfy $\rho^A = \sigma^A = \frac{1}{d_A} \mathbb{1}^A$, we can express the bipartite states as the Choi matrices $\rho^{AB} = \mathrm{id} \otimes \mathcal{D}\left(\varphi_+^{AA'}\right)$ and $\sigma^{AC} = \mathrm{id} \otimes \mathcal{F}\left(\varphi_+^{AA'}\right)$, where $\mathcal{D} : \mathcal{B}(\mathcal{H}_{A'}) \to \mathcal{B}(\mathcal{H}_B)$ and $\mathcal{F} : \mathcal{B}(\mathcal{H}_{A'}) \to \mathcal{B}(\mathcal{H}_C)$ are two quantum processes (CPTP maps), and $\phi_+^{AA'}$ is the projection on the maximally entangled state $|\phi_+^{AA'}\rangle = \frac{1}{\sqrt{d_A}} \sum_{i=1}^{d_A} |ii\rangle$, where $\{|i\rangle\}_{i=1}^{d_A}$ is an orthonormal basis for $A$. Therefore, in this case the condition $\mathrm{id} \otimes \mathcal{E}(\rho^{AB}) = \sigma^{AC}$ becomes equivalent to the degradability of $\mathcal{D}$ into $\mathcal{F}$, that is, $\mathcal{F} = \mathcal{E} \circ \mathcal{D}$, and we denote it simply by $\mathcal{F} \prec_q \mathcal{D}$.

Quantum majorization hence generalizes classical stochasticity and captures the notion that the process $\mathcal{F}$ is in some sense "more disordered" than $\mathcal{D}$, since it can be obtained from $\mathcal{D}$ via $\mathcal{E}$. However, it does not say anything about $\mathcal{E}$, which can be a completely general quantum process. Typically, in resource theories, it is important to place some additional restrictions on allowed (or "free") processes, and demand that $\mathcal{E}$ is a free operation of the theory. Many resource theories, such as

entanglement theory, do not admit a simple specification, however, as we shall see shortly, in both the resource theories of asymmetry and thermodynamics, such a restriction of $\mathcal{E}$ to lie in a subset of free (symmetric or thermodynamic) processes can be made with a natural modification of our core result.

**Characterization of quantum majorization.** Given the two bipartite states $\rho^{AB}$ and $\sigma^{AC}$, how can we determine whether $\rho^{AB}$ quantum majorizes $\sigma^{AC}$? One simple and intuitive necessary condition, that follows from the data processing inequality, is that

$$S(A|B)_\rho \leq S(A|C)_\sigma , \tag{1}$$

where $S(A|B) = S(A,B) - S(B)$ is the conditional entropy, and $S(\rho) = -Tr[\rho \log \rho]$ is the von-Neumann entropy. The intuition is that, if $\sigma^{AC} \prec_q \rho^{AB}$, then information about system $A$ is more accessible from system $B$ than from system $C$. Hence, the uncertainty of $A$ given $B$, i.e., $S(A|B)$, can only be smaller than the uncertainty of $A$ given $C$, i.e., $S(A|C)$. However, only one entropic condition is far from being sufficient to completely characterize quantum majorization.

In order to produce more necessary conditions, one can use a similar intuition to generate infinitely many necessary conditions that follows from the following observation (Fig. 1):

$$\sigma^{AC} \prec_q \rho^{AB} \Rightarrow \Phi \otimes \mathrm{id}(\sigma^{AC}) \prec_q \Phi \otimes \mathrm{id}(\rho^{AB}) \tag{2}$$

for any quantum process $\Phi : \mathcal{B}(\mathcal{H}_A) \rightarrow \mathcal{B}(\mathcal{H}_{A'})$. Note that $\Phi$ is acting on system $A$ while $\mathcal{E}$ in Definition 1 is acting on system $B$. We therefore conclude that if $\sigma^{AC} \prec_q \rho^{AB}$ then, for any quantum process $\Phi$, we must have:

$$S(A'|B)_{\Phi \otimes \mathrm{id}(\rho^{AB})} \leq S(A'|C)_{\Phi \otimes \mathrm{id}(\sigma^{AC})} . \tag{3}$$

While the conditions above are necessary, again they are not sufficient, and even in the purely classical case: there exist classical states $\rho^{AB}$ and $\sigma^{AB}$ such that $\sigma^{AC} \nprec_q \rho^{AB}$, even though the above equation holds of all $\Phi$ (and any dimensions of $A'$)[26,27].

On the other hand, in the following central result of our paper, we show that if one replaces the conditional (von-Neumann) entropy in (3) with the conditional min-entropy[28], then the inequalities in (3) indeed provide, if all simultaneously satisfied, a sufficient condition for quantum majorization. Moreover, we can restrict $\Phi$ to be an entanglement breaking channel, and bound the dimension of system $A'$ to be no greater than the dimension of system $C$. Similar results, dubbed "reverse data-processing theorems," have been obtained before[18,25,27,29], although in a different framework involving extra ancillas and a classical

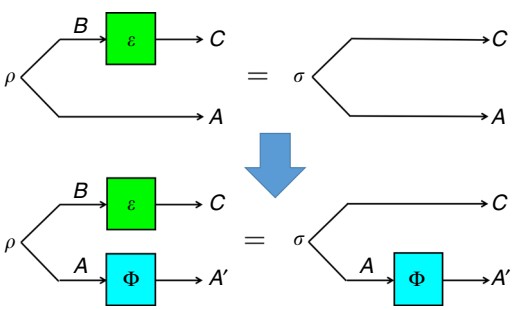

**Fig. 1** Quantum majorization. The condition of quantum majorization $\sigma^{AC} \prec_q \rho^{AB}$ implies the infinite set of relations $(\Phi \otimes \mathrm{id})(\sigma^{AC}) \prec_q (\Phi \otimes \mathrm{id})(\rho^{AB})$, where $\Phi$ is any CPTP map acting on system A (cfr. Eq. (2) in the main text). Theorem 1 provides a complete set of monotones for quantum majorization, expressed as entropic functions of the bipartite state and the channel $\Phi$ acting on it

reference system, while the present relations are fully quantum and do not need additional external systems.

The conditional min-entropy, $H_{\min}(A|B)_\Omega$, of a bipartite state $\Omega^{AB}$, is defined as[28]

$$H_{\min}(A|B)_\Omega := -\log \inf_{\tau^B \geq 0} \{Tr[\tau^B] : 1^A \otimes \tau^B \geq \Omega^{AB}\}. \tag{4}$$

It is known to be a single-shot analog of the conditional (von-Neumann) entropy. This analogy is particularly motivated by the fully quantum asymptotic equipartition property[30], which states that in the asymptotic limit of many copies of $\Omega^{AB}$, the smooth version of $H_{\min}(A|B)$ approaches the conditional (von-Neumann) entropy. The conditional min-entropy has numerous applications in single-shot quantum information (e.g., ref. [30] and references therein), quantum hypothesis testing (e.g., refs. [19,27] and references therein), and quantum resource theories[31].

Theorem 1: Let $\rho^{AB} \in \mathcal{B}(\mathcal{H}_A \otimes \mathcal{H}_B)$ and $\sigma^{AC} \in \mathcal{B}(\mathcal{H}_A \otimes \mathcal{H}_C)$ be two compatible bipartite quantum states. Let $\{M_j^A\}$ be an arbitrary, but fixed, informationally complete POVM on system $A$. Denote the dimension of any system $X$ as $d_X \in \mathbb{N}$. The following are equivalent:

1. The state $\rho^{AB}$ quantum majorizes $\sigma^{AC}$,

$$\sigma^{AC} \prec_q \rho^{AB}. \tag{5}$$

2. For any quantum process (CPTP linear map) $\Phi : \mathcal{B}(\mathcal{H}_A) \rightarrow \mathcal{B}(\mathcal{H}_{A'})$, with $d_{A'} = d_C$,

$$H_{\min}(A'|B)_{\Phi \otimes \mathrm{id}(\rho^{AB})} \leq H_{\min}(A'|C)_{\Phi \otimes \mathrm{id}(\sigma^{AC})} \tag{6}$$

3. Eq. (6) holds for any measure-and-prepare quantum channel $\Phi : \mathcal{B}(\mathcal{H}_A) \rightarrow \mathcal{B}(\mathcal{H}_{A'})$ of the form:

$$\Phi(\eta^A) = \sum_{j=1}^{d_A^2} Tr[M_j^A \eta^A] \omega_j^{A'} , \tag{7}$$

while the states $\{\omega_j^{A'}\}$ can freely vary.

4. $g(\rho^{AB}, \sigma^{AC}) \geq 1$, where the the function $g$ is defined by the following semidefinite programming:

$$g(\rho^{AB}, \sigma^{AC}) = \max$$

$$\left\{ y \mid \forall j \, y \sigma_j^T \leq Tr_B[\tau^{CB}(I \otimes \rho_j)], \ \tau^{CB} \geq 0, \ \tau^B \leq I \right\} \tag{8}$$

where

$$\rho_j \equiv \frac{Tr_A[(M_j^A \otimes 1^B)\rho^{AB}]}{Tr[M_j^A \rho^A]} \text{ and}$$

$$\sigma_j \equiv \frac{Tr_A[(M_j^A \otimes 1^C)\sigma^{AC}]}{Tr[M_j^A \sigma^A]}. \tag{9}$$

The proof of the above theorem is postponed to Supplementary Note 1.

Remark 2: In the classical case, both $\rho^{AB} = \sum_{x,y} p_{xy}|x\rangle\langle x| \otimes |y\rangle\langle y| \equiv P$ and $\sigma^{AC} = \sum_{x,z} q_{xz}|x\rangle\langle x| \otimes |z\rangle\langle z| \equiv Q$ are diagonal, where $P$ (and $Q$) is the matrix whose components are the probabilities $p_{xy}$ ($q_{xz}$). Therefore, the relation $\sigma^{AC} = \mathrm{id} \otimes \mathcal{E}(\rho^{AB})$ can be expressed as $Q = SP$, where $S$ is a column stochastic matrix, so that $Q^T \prec_m P^T$ (The relation $Q = SP$ is equivalent to $Q^T = P^T S^T$, with $S^T$ being a row stochastic matrix.). Dahl

obtained in ref. [16] that $P$ matrix-majorizes $Q$ if and only if for all sub-linear functionals $f$, that can be written as a maximum of a finite number of linear functionals, the following holds:

$$\sum_j f(\mathbf{p}_j) \geq \sum_k f(\mathbf{q}_k) , \qquad (10)$$

where $\mathbf{p}_j$ and $\mathbf{q}_k$ are the rows of $P$ and $Q$, respectively. Since classically $2^{-H_{\min}(A|B)}$ is a sub-linear functional (see more details in the Supplementary Notes 1 and 4), our theorem above provides the same result for the classical case, with a slight improvement that $f$ can be restricted to sub-linear functionals that can be written as a maximum of at most $d_C$ linear functionals.

Remark 3: The conditions in Eq. (6) are given in a form of monotones; i.e., functions that behave monotonically under certain operations (in our case under quantum majorization). In quantum resource theories monotones quantify resources as they do not increase under free operations. As we will see below, the conditional min-entropies that appear in Theorem 1 can be used to quantify asymmetry in the resource theory of quantum reference frames[32], and athermality in quantum thermodynamics. Since Eq. (6) has to hold for any CPTP map $\Phi : A \to A'$ (or measurement-prepare channels with any set of density matrices $\{\omega_j^A\}$), quantum majorization is characterized in Theorem 1 by means of an infinite number of monotones. This can be related with the fact that here we consider exact transformations, and typically an exact (algebraic) solution to such an SDP feasibility problem is NP-hard. However, part 4 of the theorem demonstrates that the question of weather or not $\rho^{AB}$ quantum majorizes $\rho^{AC}$ can be solved efficiently using semidefinite programming. A discussion comparing the two formulations, i.e., one SDP versus infinite monotones, is presented the supplementary material Note 3.

If only system $A$ is classical, that is the states $\rho^{AB} = \sum_i p_i |i\rangle\langle i| \otimes \rho_i$ and $\sigma^{AC} = \sum_i p_i |i\rangle\langle i| \otimes \sigma_i$ are classical-quantum states, we get that (5) is equivalent to

$$\sigma_i = \mathcal{E}(\rho_i) \qquad (11)$$

for all $i$ such that $p_i > 0$. This is a classic problem in quantum hypothesis testing, and the results presented here complement previous results in the same direction[22,23,25,27,33–35]. In particular, it can be shown (see Lemma 1 in the Supplementary Note 1) that Theorem 1 above implies the following corollary:

Corollary 1: There exists $\mathcal{E}$ satisfying (11) if and only if for any set of $n$ density matrices $\{\omega_i^A\}_{i=1}^n$, we have $H_{\min}(A|B)_\Omega \leq H_{\min}(A|C)_\Omega$, where

$$\Omega^{ABC} = \frac{1}{n} \sum_{i=1}^n \omega_i^A \otimes \rho_i^B \otimes \sigma_i^C . \qquad (12)$$

The same relation holds if the uniform distribution $1/n$ is replaced with any other arbitrary distribution $q_i$, with the only condition that $q_i > 0$.

**A complete set of entropic conditions for the resource theory of asymmetry.** So far we considered the relation $\sigma^{AC} = \mathrm{id} \otimes \mathcal{E}(\rho^{AB})$ with arbitrary CPTP map $\mathcal{E} : \mathcal{B}(\mathcal{H}_B) \to \mathcal{B}(\mathcal{H}_C)$. We now impose additional constraint on $\mathcal{E}$, requiring it to be $G$-covariant with respect to a compact group $G$. That is, $\mathcal{E}$ is $G$-covariant with respect to two unitary representations of $G$ on systems $B$ and $C$, denoted, respectively, by $\{V_g\}_{g \in G}$ and $\{U_g\}_{g \in G}$, if

$$U_g \mathcal{E}(\rho) U_g^{-1} = \mathcal{E}(V_g \rho V_g^{-1}) \quad \forall g \in G . \qquad (13)$$

We write $\sigma^{AC} \prec_q^G \rho^{AB}$, if $\sigma^{AC} = \mathrm{id} \otimes \mathcal{E}(\rho^{AB})$ with a $G$-covariant CPTP map $\mathcal{E}$.

Theorem 1 can be easily upgraded to accommodate $G$-covariant maps: the formal statement is given as Theorem 2 in Supplementary Note 5. Particularly, it can be shown that $\sigma^{AC} \prec_q^G \rho^{AB}$ if and only if

$$H_{\min}(A'|B)_{\mathcal{G}[\Phi \otimes \mathrm{id}(\rho^{AB})]} \leq H_{\min}(A'|C)_{\mathcal{G}[\Phi \otimes \mathrm{id}(\sigma^{AC})]} \qquad (14)$$

for all CPTP entanglement breaking maps $\Phi : \mathcal{B}(\mathcal{H}_A) \to \mathcal{B}(\mathcal{H}_{A'})$ of the form (7). Here $\mathcal{G} : \mathcal{B}(\mathcal{H}_{A'} \otimes \mathcal{H}_C) \to \mathcal{B}(\mathcal{H}_{A'} \otimes \mathcal{H}_C)$ is the bipartite $G$-twirling map given by

$$\mathcal{G}[\tau^{A'C}] = \int dg \, (\overline{U}_g \otimes U_g) \, \tau^{A'C} \, (\overline{U}_g^\dagger \otimes U_g^\dagger) , \qquad (15)$$

where the over bar denotes the complex conjugation made with respect to an arbitrary but fixed orthonormal basis.

In the special case in which both $\rho^{AB} = |0\rangle\langle 0|^A \otimes \rho^B$ and $\sigma^{AC} = |0\rangle\langle 0|^A \otimes \sigma^C$ are product states, our theorem is simplified to the following statement: $\rho^B$ can be converted to $\sigma^C$ by a $G$-covariant map if and only if for any density matrix $\eta^{A'}$,

$$H_{\min}(A'|B)_{\mathcal{G}[\eta^{A'} \otimes \rho^B]} \leq H_{\min}(A'|C)_{\mathcal{G}[\eta^{A'} \otimes \sigma^C]} . \qquad (16)$$

Therefore, the quantities $H_{\min}(A'|B)_{\mathcal{G}[\eta^{A'} \otimes \rho^B]}$, for varying reference state $\eta^{A'}$, provide a complete set of asymmetry monotones for the resource theory of asymmetry[36–40]. In other words, for any given state $\eta^{A'}$, the functions $H_{\min}(A'|B)_{\mathcal{G}[\eta^{A'} \otimes \rho^B]}$ provide a single-copy characterization of the $G$-asymmetry content of state $\rho^B$. That is to say, even though asymmetry is not a state function in itself (as it is not totally ordered), it can still be completely described in terms of a complete set of such state functions. We are now ready to discuss the application of this result to quantum thermodynamics.

**A complete set of entropic conditions for quantum thermodynamics.** While thermodynamics in macroscopic, equilibrium, and classical regimes is well understood[41], there is the fundamental question of how one can extend thermodynamic notions into non-equilibrium, finite-sized systems[42–44], and in particular systems displaying highly non-classical properties such as quantum coherence, contextuality, and entanglement[45–49]. One particular approach to this problem[3–14] has been to utilize tools and concepts developed in the study of entanglement, which is understood within the framework of resource theories. A resource theory provides a way to quantify physical characteristics that are not simply given by Hermitian observables, and is defined once we specify a set of free states, as those that do not have the properties one wishes to study, together with set of free operations, that are compatible with the set of free states in the sense that their action on any free state always yields another free state.

This approach of analyzing thermodynamics in terms of its process structure (instead of starting with problematic terms such as "heat" or "work" or "entropy") turns out to have a long and successful history dating back to the 1909 seminal work of Carathéodory[50]. Other notable accounts were obtained in 1964 by Giles[51] and more recently in 1999 by Lieb and Yngvason[52], who provided a thorough analysis in terms of adiabatic accessibility. Moreover, it has recently been shown in[13] that the thermodynamic structure of incoherent quantum states obtained from an information-theoretic perspective coincides with the phenomenological analysis in ref. [52], which demonstrates the soundness of the resource theoretic approach.

In thermodynamics, a preferred class of states are singled out as free states from the condition of complete passivity[53,54]. In the simplest case, the Gibbs state $\frac{1}{Z} e^{-\beta H}$, with $\beta = (kT)^{-1}$ and $Z = \mathrm{Tr}[e^{-\beta H}]$, is the only quantum state that can be freely admitted without trivializing the theory energetically. More

generally, in the presence of additional additive conserved charges $\{X_1, \ldots X_n\}$, such as angular momenta and particle numbers, this can be extended (under certain assumptions on external constraints[41,55–61]) to the generalized Gibbs state

$$\gamma^A = \frac{1}{\mathcal{Z}} e^{-\beta(H^A - \sum_k \mu_k X_k^A)}, \tag{17}$$

with $\{\mu_k\}$ being Lagrange multiplier constants for the conserved quantities and $\mathcal{Z} = Tr[e^{-\beta(H^A - \sum_k \mu_k X_k^A)}]$. In the case that we just have a single additional number operator $N$, the constant is the usual chemical potential[41].

**Generalized thermal processes**. Our thermodynamic framework is an extension of the resource theory of thermal operations (TOs)[4,5,7] to a set of transformations that contains TOs as a proper subset. It is an extension in two ways: firstly, it makes a weaker assumption about the underlying microscopic process, and secondly it is defined in terms of a collection of distinguished thermodynamic observables, such as those in the generalized Gibbs ensemble, and not just in terms of energy. We elaborate on the relation between the two classes in Supplementary Note 8. We shall refer to these free transformations as generalized thermal processes (abbreviated to TPs), and they are specified by the following three physical assumptions:

- *A1. Microscopic conservation:* Each input quantum system and output quantum system has a Hamiltonian $H$, and a collection of distinguished observables $X_1, \ldots X_n$. The total energy and the observables $\{X_k\}$ are conserved microscopically in any free process, and moreover $[H, X_k] = 0$ for all $k = 1, \ldots, n$.
- *A2. Equilibrium preservation:* For every (input or output) system $A$, an equilibrium free state exists that is stable under the class of free processes.
- *A3. Incoherence:* The free processes do not require any sources of quantum coherence between eigenbases of conserved quantities.

The microscopic conservation assumption ensures that every quantum system $A$ has a well-defined Hamiltonian $H^A$ at the initial time and some other Hamiltonian $H^{A'}$ at the final time. It also allows for an arbitrary set of additional conserved charges, as discussed. More precisely, any TP map $\mathcal{E}$ on $A$ admits a Stinespring dilation onto some larger system $B$ such that

$$\mathcal{E}(\rho^A) = Tr_C V(\rho^A \otimes \sigma^B) V^\dagger \tag{18}$$

where $B$ is some other quantum system defining the thermal environment. The microscopic conservation assumption implies that the isometry $V$ obeys

$$V(H^A \otimes 1^B + 1^A \otimes H^B) = (H^{A'} \otimes 1^C + 1^{A'} \otimes H^C)V$$

$$V(X_k^A \otimes 1^B + 1^A \otimes X_k^B) = (X_k^{A'} \otimes 1^C + 1^{A'} \otimes X_k^C)V \tag{19}$$

for all $k = 1, \ldots n$, which defines the microscopic energy conservation and the conservation of the charges. Note that we also allow the input system and output to differ, which may occur due to the presence of strong-couplings that affect factorizability into independent subsystems. It is also important to emphasize that we do not assume or require microscopic control of $V$. It is only the total process $\mathcal{E}$ that is experimentally relevant. The particular set of observables are determined by the physical context and we shall refer to them as the thermodynamic observables for the system.

The equilibrium preservation assumption says that for every system $A$ there is a state $\rho_*^A$, such that $\mathcal{E}(\rho_*^A) = \rho_*^A$ for all TPs $\mathcal{E}$. However (A1) singles out a set of distinguished observables $\{H^A, X_1^A, \ldots, X_n^A\}$ that microscopically are additively conserved. The fact that $\rho_*^A$ is a free state of the theory implies[58–60] that the only form of $\rho_*^A$ that can yield a non-trivial resource theory in these observables is one for which $\log \rho_*^A$ is a linear combination of the observables—namely it must be a generalized Gibbs state $\gamma^A$ as defined in (17), at some fixed temperature $T = (k\beta)^{-1}$ and Lagrange multipliers $\mu_1, \ldots, \mu_n$. Therefore the free states of the theory are defined uniquely by these parameters.

The final assumption on incoherence is a statement of non-classicality within the theory and requires us to provide an explicit accounting for coherence resources. It is known for thermal operations that if the only coherences present are within energy eigenspaces then the resultant theory is essentially classical, and is described by thermo-majorization[5]. However, coherences between energy eigenspaces behave differently and do not have such a classical description[10]. Therefore one must carefully account for these coherences thermodynamically. The precise formulation of this requirement in the case of energy is that if any free process $\mathcal{E}$ is discussed in Supplementary Note 6, and has the consequence that any $\mathcal{E}$ can be represented as

$$\mathcal{E}(\rho) = Tr_C V(\rho^A \otimes \sigma^B) V^\dagger \tag{20}$$

where $V$ is a conserving interaction, and $\sigma^B$ is an external state incoherent in the energy eigenbases and obeys $\sigma^B = \lim_{\tau \to \infty} \frac{1}{\tau} \int_0^\tau dt U_B(t) \sigma^B U_B^\dagger(t)$. This captures the notion that $\mathcal{E}$ is realized without consuming any coherent resources from the external degrees of freedom in $B$. At the level of quantum operations on $S$, this implies that we have the following symmetry property for all free operations

$$U'(t)\mathcal{E}(\rho^A)U'(t)^\dagger = \mathcal{E}(U(t)\rho^A U(t)^\dagger) \tag{21}$$

where $U(t) = \exp[-itH^A]$ and $U'(t) = \exp[-itH^{A'}]$ are respectively free evolution of the input/output system for an interval of time $t$. The operation $\mathcal{E}$ is said to be covariant under time-translation. The more general case of multiple conserved charges is discussed below.

The three physical assumptions specify the set of generalized thermal processes, and it is readily seen that it contains the set of thermal operations. In the case when the only conserved quantity is $H$, there is no particular physical reason to choose one set of operations over the other. However, in the case of multiple conserved charges $X_1, \ldots, X_n$, the use of TPs has an advantage in that it allows one to handle generalized Gibbs ensemble scenarios more easily. The details of system $B$ are, in general, not observed thermodynamical degrees of freedom, and with an explicit microscopic specification, such as with thermal operations, subtleties arise in the case of additional charges. Particularly, subtleties arise if one wishes to have non-trivial $\mu_k$ Lagrange multipliers in the generalized Gibbs ensemble (17) and also satisfy the microscopic conservation assumption. The formulation here simply avoids this by not demanding a specific form for the microscopic state $\tilde\sigma^B$ in the definition of the free processes. The incoherence assumption only constrains the microscopic details to the extent that there are no observable effects of coherence at the level of the process $\mathcal{E}$.

In the Supplementary Note 7, we show that our core result on quantum majorization can be adapted to the setting of generalized thermal processes to fully describe the state interconversion structure. This is obtained by establishing the following lemma, which is proved in the Supplementary Note 6.

Lemma 1: Consider two sets of thermodynamic observables $\{H^S, X_1^S, \ldots, X_n^S\}$ for quantum system $S = A$ and quantum

systems $S = A'$. Then, the set of all quantum processes from $A$ into $A'$ defined by (A1-A3) coincides with the set of all $\gamma$-preserving processes on $A$ that are covariant under the group $G$ generated by the thermodynamic observables on $A$ and $A'$.

**State conversions under thermal processes**. Since TPs are $G$-covariant we may make use of our earlier results on $G$-covariant state interconversion of a collection of states $\{\rho_i^A\}$ into $\{\sigma_i^B\}$. We first consider the case where energy is the only distinguished thermodynamic observable that is conserved microscopically. Combining the $G$-covariant version of Theorem 1 with the above lemma we get the following theorem (see Supplementary Note 7 for more details).

Theorem 2: Let $A$ and $A'$ be two quantum systems, with the respective Hamiltonians $H^A$ and $H^{A'}$ being the only thermodynamic observables, and let $0 < q < 1$ be an arbitrary but fixed number. The state transformation $\rho^A \to \sigma^{A'}$ is possible under generalized thermal processes at a temperature $(k_B\beta)^{-1}$ if and only if for all reference frame systems $R$ with the same dimension as of $A'$ and with Hamiltonian $H^R = -(H^{A'})^T$, and for all pairs of states $\boldsymbol{\eta} = (\eta_1^R, \eta_2^R)$, we have

$$S_{\boldsymbol{\eta}}(\rho^A) \leq S_{\boldsymbol{\eta}}(\sigma^{A'}) , \tag{22}$$

where $S_{\boldsymbol{\eta}}(\rho^A) := H_{\min}(R|A)_\Omega$ and

$$\Omega^{RA} = \langle q\eta_1^R \otimes \rho^A + (1-q)\eta_2^R \otimes \gamma^A \rangle . \tag{23}$$

Here, $\gamma^A = \exp[-\beta H^A]/Z$ is the Gibbs state on $A$, and $\langle \omega^{RA} \rangle \equiv \lim_{\tau \to \infty} 1/\tau \int_0^\tau dt\, U(t)\omega^{RA}U^\dagger(t)$ is the channel that maps any state of $RA$ to its time-averaged version, and $U(t) = \exp[-it(H^R \otimes 1^A + 1^R \otimes H^A)]$ is the unitary time-evolution under the Hamiltonian for the composite system $RA$.

It is important to note that these conditions can be greatly reduced. In particular one can simply consider $q = \frac{1}{2}$ alone, however, in some cases it is useful to choose different values and so we give the general case here. Also, it readily seen that the state $\eta_2^R$ can be chosen to be block-diagonal in the energy eigenbasis, while $\eta_1^R$ can be restricted to reference frame states that have the same modes of coherence as $\rho^A$[12,37].

**Time-energy constraints on state conversions**. Next, we show that the necessary and sufficient condition found in Theorem 2 has an interesting physical interpretation; loosely speaking, it implies that a state conversion is possible in quantum thermodynamics, if and only if it does not lead to any net increase in work or time-information.

A key obstacle in quantum thermodynamics is that to determine the existence of the transformation $\rho^A \to \sigma^{A'}$, one needs to consider two different types of physical properties of states: (i) properties related to their energy distribution, which leads to conditions such as thermo-majorization[62], and (ii) properties related to the coherence in the energy eigen-basis. Roughly speaking, one needs to check that the initial state $\rho^A$ has (at least) as much as free energy and coherence as the desired final state $\sigma^{A'}$.

It is not possible in general to quantify both of these simultaneously in a measurement scheme. Coherences in energy are precisely the time-dependent components of a quantum system and thus one encounters an obstacle of complementarity between time and energy measurements. Physically these two aspects can be viewed as "clock" and "work" regimes of a quantum system. Theorem 2 gets around this complementarity by allowing the reference system $R$ to act simultaneously as a "clock/work reference". In other words, one can interpolate

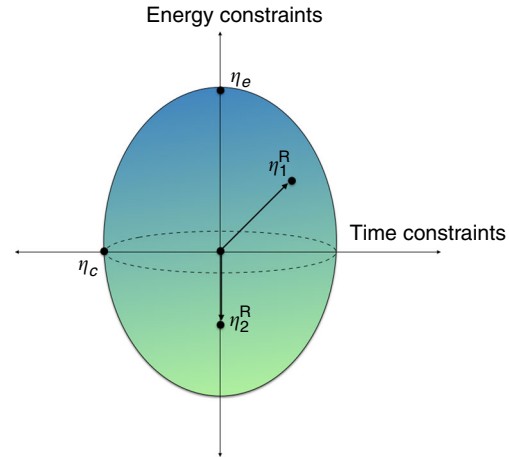

**Fig. 2** Time-energy constraints for thermal processes. The entropic conditions for a state transformation $\rho^A \to \sigma^{A'}$ under TPs are defined with respect to a quantum reference frame $R$ and two states $\eta_1^R$ and $\eta_2^R$. The schematic vertical axis denotes states block-diagonal in energy (e.g., an energy eigenstate $\eta_e = |E\rangle\langle E|$), while the horizontal axis denotes states with maximal time-dependent oscillations—'clock' states $\eta_C$ of $R$. When $\eta_1^R$ is confined to being incoherent (the vertical axis) we recover thermo-majorization. For $R$ being macroscopic and $\eta_1^R = \eta_C$, we obtain a Page–Wootters clock constraint on the thermodynamic transformation. Varying $\eta_1^R$ smoothly interpolates between the time constraints and energy constraints

smoothly between the two regimes via the different choices of quantum states $\eta^R$. This is illustrated schematically in Fig. 2.

To see this better, we first consider the case where either the input or output state is incoherent in the energy eigenbasis. This regime is described by an essentially classical stochastic energy condition. The following result is shown in the Supplementary Note 4.

Corollary 2: Let $A$ and $A'$ be two quantum systems, with respective Hamiltonians $H^A$ and $H^{A'}$ being the only thermodynamic observables. Let $\rho^A$ and $\sigma^{A'}$ be quantum states on the input and output systems, respectively. If either $[\rho^A, H^A] = 0$ or $[\sigma^{A'}, H^{A'}] = 0$, then the state transformation $\rho^A \to \sigma^{A'}$ is possible under generalized thermal processes at a temperature $T = (k\beta)^{-1}$ if and only if $[\sigma^{A'}, H^{A'}] = 0$ and $\rho^A$ thermo-majorizes $\sigma^{A'}$.

This recovers previous results[5] on quantum thermodynamics for the case of one of the states having no coherences between energy eigenspaces. Moreover, in the case of incoherent input $\rho^A$, the use of a coherent reference state $\eta^R$ does not yield any additional constraint. Specifically, $\Omega^{RA} = q\langle\eta_1^R\rangle \otimes \rho^A + (1-q)\langle\eta_2^R\rangle \otimes \gamma^A$, and so the coherence of states $\eta_1^R$ and $\eta_2^R$ is irrelevant. The only relevant constraints in state transformation $\rho^A \to \sigma^{A'}$ are constraints related to the energy distribution of states.

On the other hand, if both the input-output states $\rho^A$ and $\sigma^{A'}$ contain coherence, then by choosing reference states $\eta_1^R$ which contain coherence, we obtain new additional coherence constraints, i.e., constraints independent of thermo-majorization. Note that coherence with respect to energy eigenbasis is equivalent to symmetry-breaking (asymmetry) with respect to time-translations generated by the system Hamiltonian. In other words, coherence of states $\rho^A$ and $\sigma^{A'}$ is related to how well time $t$ can be estimated from states $\rho^A(t) = e^{-iH_A t}\rho^A e^{iH_A t}$ and $\sigma^{A'}(t) = e^{-iH_{A'} t}\sigma^{A'} e^{iH_{A'} t}$.

The TPs are both covariant under time-translation and preserve the Gibbs state. In the Supplementary Note 6, we will show that the converse is also true (i.e. a covariant Gibbs preserving map is a TP). Therefore, previously discussed

measures, such as those that are based on Renyi divergences of the form $A_\alpha(\rho^A) = S_\alpha(\rho^A||\langle\rho^A\rangle)$, behaves monotonically under TPs, and provide independent thermodynamic constraints beyond thermo-majorization[10]. One can also use constraints on modes of coherence[12,37] and the Fisher Information[63], to derive other independent measures of athermality. However, the set of conditional min-entropy measures obtained here is complete and therefore sufficient to imply the monotonicity of all of these measures.

In the Supplementary Note 4, we show that the entropic conditions with $\eta_1^R$ being incoherent in energy leads to thermo-majorization, and captures the degree to which the system $A$ is ordered in energy. Since in quantum systems one has complementarity between time and energy one might expect that the case of $\eta_1^R$ being highly coherent in energy might therefore capture the degree to which $A$ is ordered in some temporal sense.

This turns out to be the case, although since time forms a continuous one-parameter group there are technical obstacles to making this statement precise. However, as we show in the Supplementary Note 9, one can in general make finite precision approximations and model time evolution for any finite dimensional quantum system (which can be assumed to have an energy spectrum of rational numbers and thus has periodic dynamics under its Hamiltonian) with the discrete group $\mathbb{Z}_N$, for some sufficiently large $N$ and with $t = n\varepsilon$. Here $\varepsilon > 0$ is the minimal time interval that can be resolved. The representation of this discrete group on $A$ is given by $n \mapsto U_\varepsilon^A(n) := \exp[-in\varepsilon H^A]$ and so the system is modeled as evolving in discrete time steps. Under these approximations, one can replace microscopic conservation assumption with a slightly weaker version described in Supplementary Note 9, and the interconversion conditions can be repeated for $G = \mathbb{Z}_N$ instead.

We define clock-times as the discrete instances $t = 0, \varepsilon, \ldots, n\varepsilon, \ldots, (N-1)\varepsilon$ for the joint system $\mathcal{H}_R \otimes \mathcal{H}_A$. As shown in the Supplementary Note 10, there exist reference frame systems $R$ that can provide a perfect classical encoding of the clock times into quantum states $\{|0^R\rangle, |1^R\rangle, \ldots, |N-1^R\rangle\}$, and for which $U_\varepsilon^R(n)|0^R\rangle = |n^R\rangle$ for any $n$. Moreover, these clock states are built from uniform superpositions in the energy eigenstates of $R$, and so are in a sense "maximally" coherent in energy. Given this, we can now demonstrate the claimed complementarity between time and energy and how it relates to the state of the reference $R$. We choose $\eta_1^R = |0\rangle^R$ and consider the limit $q \to 1$, which corresponds to the condition of time-translation covariance alone. For this one can show that

$$\Omega^{RA} = \frac{1}{N}\sum_{k=0}^{N-1}|k\rangle\langle k|^R \otimes \rho^A(n), \qquad (24)$$

where $\rho^A(n) := U_\varepsilon^A(n)\rho^A(U_\varepsilon^A(n))^\dagger$ is the state of the system $A$ at the $n$th clock time for the joint system. Now, since $\Omega^{RA}$ is a classical-quantum state, we have that[64,65]

$$H_{\min}(R|A)_\Omega = -\log p_{\text{guess}}, \qquad (25)$$

where $p_{\text{guess}}$ is the optimal Helstrom guessing probability for the ensemble of states $\{(\frac{1}{N}, \rho^A(n))\}_{n=0}^{N-1}$ on $A$. This implies that $2^{-H_{\min}(R|A)_\Omega}$ is the optimal guessing probability of the clock time $t = n\varepsilon$ for the joint system, given the single copy of $\rho^A$. Monotonicity of $H_{\min}(R|A)_\Omega$ under the thermal processes therefore implies monotonicity of the clock time guessing probability for the system. Phrased differently, the time-translation covariance property of thermal processes implies that the ability of the thermodynamic system $A$ to act a quantum clock[66] can never increase. This demonstrates how the reference frame system $R$ functions to define both time and energy constraints on the state interconversion for the system $A$.

We note that this result connects with foundational work by Page and Wootters[67], who considered how one can have dynamics in a universe that is covariant in time. They proposed a conditional probability formalism, which mirrors our present set up and relies on covariant measurements with $P(X^R = x|Y^A = y)$, the probability that some observable $X^R$ has a sharp value given a measurement of $Y^A$ yielding a particular result. These relational expressions were shown to describe dynamics within the time-translation invariant global state, such as $\Omega^{RA}$ here.

We note that the condition in Theorem 2 that $H^R = -(H^A)^T$ can be understood in the context of global time-translation covariance. Formally, it says that the free evolution of the reference system $R$ is via the representation dual to the time-translation action on $A$. More physically, it says that the joint Hamiltonian $H^{RA} = H^R \otimes 1^A + 1^R \otimes H^A$ admits a non-trivial eigenspace with zero total energy in which coherent dynamics on $A$ can be fully encoded. This should be compared with the Wheeler de Witt equation $H_{\text{tot}}|\Psi\rangle = 0$ in quantum gravity, which also ensures global covariance under time-translations for the universe[68].

**Multiple conserved charges.** Finally, we can state the necessary and sufficient conditions for the case of having additional, additively conserved observables $\{X_1, \ldots, X_n\}$. In this case assumption (A3) follows a similar argument to the one for energy, and the auxiliary system can be assumed to be in a state $\sigma^B$ for which $\sigma^B = e^{-isX_k^B}\sigma^B e^{isX_k^B}$, for all $s \in \mathbb{R}$ and for any thermodynamic observable $X_k^B$. Ranging over all the observables, this condition can be expressed more compactly as $\sigma^B = U(g)\sigma^B U^\dagger(g)$, for all unitary transformations $U(g)$ in the Lie group $G$ generated by the observables $\{H^B, X_1^B, \ldots, X_n^B\}$. Note that this condition is equivalent to $\sigma^B = \int_G dg\, U(g)\sigma^B U^\dagger(g)$, where $dg$ is the uniform (Haar) measure over this group. Therefore, this assumption, together with (A1) imply that the process is covariant with respect to group $G$, i.e., $\mathcal{U}_g \circ \mathcal{E} = \mathcal{E} \circ \mathcal{U}_g$, where $\mathcal{U}_g(\rho^A) := U(g)\rho^A U^\dagger(g)$. In other words, the process is covariant under the symmetry group action generated by the thermodynamic observables on the input/output systems. Our main result on the thermodynamic structure of states under TPs is as follows.

Theorem 3: [Generalized thermal processes] Let $A$ and $A'$ be two quantum systems, with thermodynamic observables $\{H^A, X_1^A \ldots, X_n^A\}$ and $\{H^{A'}, X_1^{A'}, \ldots, X_n^{A'}\}$, respectively, and fix $0 < q < 1$. The state transformation $\rho^A \to \sigma^{A'}$ is possible under generalized thermal processes at a temperature $(k_B\beta)^{-1}$ and at fixed Lagrange multipliers $\mu_1, \ldots, \mu_n$, if and only if for all reference frame systems $R$ of equal dimension to $A'$ with thermodynamic observables $H^R = -(H^{A'})^T$ and $\{X_k^R = -(X_k^{A'})^T\}_{k=1}^n$, and for all pairs of states $\boldsymbol{\eta} = (\eta_1, \eta_2)$ we have $S_\eta(\rho^A) \leq S_\eta(\sigma^{A'})$, where $S_\beta(\rho^A) := H_{\min}(R|A)_\Omega$ and

$$\Omega^{RA} = \int_G dg\, U(g)(q\eta_1^R \otimes \rho^A + (1-q)\eta_2^R \otimes \gamma^A)U(g)^\dagger \quad (26)$$

where $\{U(g)\}$ is the symmetry group generated by the additively conserved observables $\{H^R \otimes 1^A + 1^R \otimes H^A, X_k^R \otimes 1^A + 1^R \otimes X_k^A; k = 1, \ldots, n\}$ on the composite system $RA$, with group parameters $g$, and $\gamma^A = \exp[-\beta(H^A - \sum_k \mu_k X_k^A)]/\mathcal{Z}$, being the generalized Gibbs ensemble on $A$.

This result is a fully covariant statement that is based on minimal assumptions, namely microscopic conservation, equilibrium preservation and incoherence, which reduces to Theorem 2 in the case of no additional thermodynamic observables beyond the system's energy.

## Discussion

In this work, we have considered a generalization of majorization for quantum processes, found a necessary and sufficient condition for this notion of majorization in terms of entropic quantities, and demonstrated some of its applications in the context of the resource theories of asymmetry and quantum thermodynamics. In particular, we derived a complete set of entropic conditions for state transformations in both of these resource theories. In contrast to the previous results, which are only applicable to restricted families of states (such as incoherent states) our approach can be applied to all states. Furthermore, these results can be generalized to the case of approximate transformations in which we only require transformations up to an epsilon smoothing. However, the approximate case requires additional tools and is left for future work.

Since our entropic monotones provide a full characterization of the resource, it is interesting to study their operational interpretations. We discussed some of these interpretations in the context of clocks. Another possible interpretation could be provided by the results of ref. [69], which relates the smoothed entropy $H_{\max}^{\varepsilon}(R|A')$ to the minimal work cost to perform a quantum process. The duality relation between min and max entropies tells us that, where $C$ purifies the state on $RA'$, and so this suggests a potential interpretation of our results in terms of generalized work costs on a purifying environment.

We also introduced a new framework for quantum thermodynamics based on the notion of generalized thermal processes, which extends thermal operations, and is based on natural physical principles. This explicitly handles coherences and is the first framework of its kind for which a complete set of state conditions has been derived.

### Data availability

The authors declare that all the data supporting the findings of this study are within the paper and its supplementary information files.

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

## Acknowledgements

G.G. would like to thank Rob Spekkens for pointing out ref. [34], and for many discussions on classical matrix majorization. We would also like to thank Eric Chitambar, Philippe Faist, Mark Girard, Kamil Korzekwa, and Matteo Lostaglio for useful discussions, and to Nicole Yunger Halpern for many comments on the first version. We also thanks for an anonymous referee who provided the simplified proof for Lemma 5 of the current version. F.B. acknowledges support from JSPS KAKENHI, Grant nos. 26247016 and 17K17796. G.G. acknowledges support from the Natural Sciences and Engineering Research Council of Canada (NSERC), D.J. is funded by the Royal Society.

## Author contributions

G.G. and D.J. conceived and developed the presented idea. Some of these ideas were conceived independently by F.B. who later on joined the project and helped both in developing the theory further, and in the writing of the paper. R.D. conceived the idea that the problem of quantum majorization can be solved with semidefinite programming and helped with the initial computations. I.M. helped with many discussions and with the revisions of the manuscript. All authors discussed the results and contributed to the final manuscript.

## Additional information

**Competing interests:** The authors declare no competing interests.

