## [Peer Review File · Nature Communications]

REVIEWERS' COMMENTS:

Reviewer #1 (Remarks to the Author):

I am satisfied with all the additions, changes and replies that the authors have made in response to my suggestions, and I can only confirm and further emphasize my previous recommendation to publish this manuscript. The characterization of the necessary and sufficient conditions for coherent transformations has been a central question for quantum thermodynamics. The fact that this manuscript achieves such a characterization, and in addition establishes a more generally applicable machinery, must be regarded as a very significant contribution to the field, and does in my view clearly justify publication in Nature Communications.

The manuscript is in a very good shape, and it can be published as it is, although I happened to notice one detail that the authors may want to change:

* Fig. 2: I guess that η_1 and η_2 in the figure should be η_1^R and η_2^R

Reviewer #3 (Remarks to the Author):

I thank the authors for their replies to my comments. I am happy to recommend this paper for publication in Nature Communications. I have only a small number of comments for the authors:

1. In reply to point 1 the authors say "Indeed we do not claim any novelty for the mathematical definition". However, the paper is still ambiguous in this respect. The beginning of Section II starts "Our generalization of matrix majorization" "which we call quantum majorization". I am not trying to be facetious, but such wording gives the impression contrary to the reply of the authors that they make no claim to novelty.

2. The authors say in their reply that "one of the main conclusions of this study in quantum thermodynamics can be precisely understood as those evolutions that can never lead to any net increase of information about time or energy". I believe the paper would benefit if something along these lines appeared in the abstract, where currently one cannot find such a clean statement. I also would recommend highlighting this "main conclusion" within the current section it is within - I think the paper would benefit if the reader could find these important messages more easily within the text.

3. The authors decided to ignore my suggestion of giving a single example of how to apply their results to any simple situation. I still find this disappointing, and believe it would only help to make the results more accessible.

Reply to Referee #1:

We thank the referee for the positive comments and for noticing the typo. We have corrected the typo.

Reply to Referee #3:

We also thank referee 3 for the comments.

Referee #3: *1. In reply to point 1 the authors say "Indeed we do not claim any novelty for the mathematical definition". However, the paper is still ambiguous in this respect. The beginning of Section II starts "Our generalization of matrix majorization" "which we call quantum majorization". I am not trying to be facetious, but such wording gives the impression contrary to the reply of the authors that they make no claim to novelty.*

Authors: To address this issue, we added the following sentence above definition 1: "Notice that notions equivalent to quantum majorization have previously been considered in Refs. [35-39] in the contexts of quantum statistics and quantum information theory."

Referee #3: 2. The authors say in their reply that "one of the main conclusions of this study in quantum thermodynamics can be precisely understood as those evolutions that can never lead to any net increase of information about time or energy". I believe the paper would benefit if something along these lines appeared in the abstract, where currently one cannot find such a clean statement. I also would recommend highlighting this "main conclusion" within the current section it is within - I think the paper would benefit if the reader could find these important messages more easily within the text.

Authors: In the previous version, this conclusion was discussed after theorem 2. Based on referee's recommendation, and to highlight this conclusion, we separated this discussion from theorem 2 by adding a subsection header, titled Time-Energy constraints on state conversions. We also added the following paragraph at the beginning of this new subsection:

Next, we show that the necessary and sufficient condition found in theorem 2 has an interesting physical interpretation; loosely speaking, it implies that a state conversion is possible in quantum thermodynamics, if and only if it does not lead to any net increase in work or time-information.

Unfortunately, due to the space limits we cannot add this conclusion to the abstract (The length of the abstract was already over the limit allowed by Nature Communications and we had to make it shorter).

Referee #3: The authors decided to ignore my suggestion of giving a single example of how to apply their results to any simple situation. I still find this disappointing, and believe it would only help to make the results more accessible.

Authors: We thank the referee for this suggestion. Inspired by the referee's suggestion, in supplementary note 5, page 11 of Supplementary Material, we added a new subsection, titled "SDP solution for thermomajorization with coherence". In this subsection, we show how our results can be applied to an example of state transformations (without charges) and we find the precise recipe for determining state transformations under Gibbs-preserving symmetric operations. We present the result in a form that can be directly plugged into CVX package of Matlab or any other standard SDP packages. Applying this recipe to special examples, reveals interesting new aspects of the resource theory of quantum thermodynamics, which will be discussed in a follow-up paper.